# Correlation between Elderly Migrants’ Needs and Environmental Adaptability: A Discussion Based on Human Urbanization Features

**DOI:** 10.3390/ijerph18105068

**Published:** 2021-05-11

**Authors:** Yi Hua, Zhi Qiu, Wenjing Luo, Yue Wang, Zhu Wang

**Affiliations:** Institute of Architectural Design and Theoretical Research, Zhejiang University, Hangzhou 310000, China; 11612015@zju.edu.cn (Y.H.); 11612013@zju.edu.cn (W.L.); 12012013@zju.edu.cn (Y.W.); 13325716901@163.com (Z.W.)

**Keywords:** elderly migrant, human urbanization feature, living need, production need, environmental adaptability

## Abstract

Building concentrated resettlement community in small towns is mostly used to deal with resettlement construction for rural migrants in economically developed regions in China, which leads to migrants’ living environment changing from rural settlements where production and living are intertwined to an urban community that only supports living functions. However, the urbanized environment is contrary to elderly migrants’ behavior, resulting in contradictions or conflicts between migrants and resettlement communities, reflecting a lack of urbanization synchronization between migrants and resettlement community environments. Further, elderly migrants are also equipped with different degrees and types of urbanization characteristics, thus reflecting different abilities to adapt to the urban community environment. Based on the corresponding relationship between people’s different production and living needs and urbanization, this research starts by investigating the production and living needs of elderly migrants, and further clarifies the environmental adaptability of elderly migrants by sorting the types and characteristics of urbanization of elderly migrants to provide a reference basis for the planning and construction of future resettlement areas. The research uses questionnaires and semi-structured interviews to investigate the population attributes and characteristics of elderly migrants, as well as their different needs for production and living. The research uses hierarchical cluster analysis, the one-way ANOVA test and Chi-square test to constructed a four-quadrant model on human urbanization features: an Urban Group with both living and production urbanized (Group H-H); a Half-urban-half-rural Group with only living needs urbanized (Group H-L); a Half-urban-Half-rural Group with only production needs urbanized (Group L-H); and a Rural group with both living and production needs not urbanized (Group L-L). Finally, based on the results, this research proposed three elderly environment construction orientations of “Promote the Supply Level of Urban Public Services”, “Create a Place That Embodies the Spirit of Immigrants’ Homeland”, and “Moderate Consideration of Agricultural Production Needs” for residential planning.

## 1. Introduction

Facing development-forced displacement and resettlement brought about by project construction and ecological exploitation during the new urbanization period, China has conducted diverse forms of urbanization resettlement practices in areas having different economic development levels [1,2,3,4]. For example, its economically developed regions, represented by the Yangtze River Delta, mostly build urban-mode resettlement communities in small towns because of high urbanization and land scarcity. There, China promotes non-agricultural production resettlement for migrants through technical training or industrial park construction due to loss of land. Migrants’ living space is transformed from villages on rural collective land to resettlement communities on urban residential land, where public services follow urban community standards, while non-agricultural production activities are located outside resettlement areas. As a result, resettlement communities support only living activities. This resettlement method has already reached the “environment” of urbanization in terms of natural land, planning modes, spatial patterns, and preset production methods.

However, due to elderly migrants’ production and living habits, rural traditions, and high degree of aging, there is a disconnect between the urbanized environment and elderly migrants’ behavior. Firstly, elderly migrants are mainly engaged in primitive agriculture before relocation, which makes elderly migrants lack experience and skills to earn a living in towns. Secondly, the village spatial structure presents organically intertwined production- and living-functions, leading to a huge difference in environmental perception and usage patterns between elderly migrants and urban residents [5,6,7]. Some studies have found a widespread “adapted use” phenomenon in resettlement communities, meaning living space is “occupied and transformed by rural-like living activities” and “requisitioned for production activities,” resulting in contradictions or conflicts between migrants’ needs and resettlement communities’ planning and control [8]. This runs contrary to the principle of “people-centered” development, reflecting a lack of urbanization synchronization between migrants and resettlement community environments.

Focusing on the problem that the environment is not synchronized with human beings in urbanization, existing studies are mostly based on the environment and tend to change people’s production and lifestyles to adapt to urbanization environment transformation, for instance, to cultural adaptation [9,10], social integration [11], ecological dependence [12], and so on. With popularization of the people-centered concept in Chinese society, studies have gradually turned to achieving mutual coordination between the environment and people by regulating the “environment”, in this case, through spatial production theory that explains processes and mechanisms of how external factors such as power, capital, and social class influence the environment. However, environmental-behavior theory posits that people’s cognitive characteristics, as an internal factor, play a more important role in environmental regulation [13,14]. In the process of Chinese urbanization, the object of this study—elderly migrants—is also evolving gradually because individual features lead to different abilities to adapt to the urban community environment, thus reflecting different degrees and types of urbanization characteristics. Therefore, clarifying correlation between cognitive characteristics and elderly migrants’ environmental adaptability is necessary, when taking human beings’ urbanization characteristics as a pre-condition of resettlement community planning and providing a theoretical basis for a new “people-centered” urbanization resettlement strategy.

## 2. Literature Review and Research Methods

### 2.1. Extraction of Urbanization Indicators for Elderly Migrants

Different from the view taking the mutual coordination between human and environment as the standard of one’s environmental adaptability in classic environmental behavior research, in the urbanization development background with strong top-down government intervention in China, the environment is regarded as a fixed feature, and the degree of one affected by environment, or the degree of urbanization, reflects one’s environment adaptability [15]. Academic research on human urbanization covers many fields, but traditional studies have focused on the household registration (hukou) system and the urban status of migrant people [16]. In recent years, the issue of human cognitive differences has also gradually gained attention [17]; for example, public health service utilization [18], educational assortative mating in marriage [19], and risk information-seeking behavior [20]. Through a literature review, this study found that elderly migrants reflect various cognitive characteristics of production and living needs. However, current studies on elderly migrants’ needs focus mainly on certain aspects of living, such as medical care and leisure, but take production participation as only a content of life experience [21]. This study attends not only to living needs reflected by “encroachment and transformation of living space in rural areas” but also to production needs reflected by “directional expropriation of living space for production activities [8].” Therefore, a dual-dimensional demand system of production and living is constructed as an evaluation criterion to measure elderly migrants’ characteristics of urbanization.

For living needs, the Chinese government has established a rights-protection system for the elderly through the Law on Protection of the Elderly’s Rights and Interests, which emphasizes living care, health care, lifelong education, social participation, and spiritual support [22]. From the perspective of equity, urbanization resettlement guarantees elderly migrants’ rights to access public services, for example, basic medical care and livelihood security [23,24]. With improved living standards, some elderly migrants have changed their lifestyle to “urban elderly” and begun to pursue all aspects of high-quality public services [25,26]. In addition, elderly migrants also seek spiritual satisfaction in collective activities and neighborhood interaction through rural traditions [27,28]. Therefore, this paper proposes “health care,” “living care,” “cultural entertainment,” and “spiritual consolation” to construct a living needs system for elderly migrants.

On production needs, compared with a single-agriculture model in their original villages, elderly migrants in resettlement communities adopt a compound-production model based on agriculture but supplemented by handicrafts, self-employment, labor hire, and other industrial and commercial activities. Although urbanization promotes transfer of rural surplus labor to secondary and tertiary industries [29], elderly migrants universally lack urban livelihood skills due to limitations on working abilities and low educational levels [30,31]. This means that only a few elderly migrants can increase their personal incomes through industrial and commercial activities, while others are still accustomed to agricultural production methods, resulting in frequent agricultural use of public space in resettlement communities. Therefore, this paper proposes “agriculture” as well as “handmade and business” to describe elderly migrants’ production-needs system.

### 2.2. Evaluation and Investigation of Environmental Adaptability

#### 2.2.1. Sample

This study intended to find a resettlement community where migrants’ adaptation and transformation to an urban environment have stabilized. In the Reservoir-L project in Zhejiang Province, the government adopted centralized resettlement for surrounding farmers, selected residential land in Town-D, and built Community-L in 2007. Community-L is constructed as an apartment block combined with some shops on side streets [8,32]. Migrants’ adaptation to the urban environment tends to be stable, and various rural behaviors are presented, thus meeting this study’s requirements. For evaluation of the degree of human urbanization, a questionnaire was designed to collect data on production and living needs, with 24 indicators in six dimensions (Table 1). Each indicator was evaluated using a 5-point Likert scale, which is widely used in social surveys to understand the degree of agreement or disagreement with a set of statements related to the measurement subject, with the degree of needs gradually increasing from 1 to 5 [33].

In China, most people are considered elderly at 55 years old, influenced by women’s retirement policy, so this study conducted a questionnaire survey for those 55 and over. In addition, other research has found that age, gender, health status, financial status, marital status, labor ability, and educational level will influence the environmental adaptability [34,35]. Therefore, this study proposes eight dimensions: age, gender, physical condition, pension, housing condition, marital status, work capacity, and educational level, to evaluate the demographic characteristics of the interviewees.

#### 2.2.2. Investigation Process

The official data shows that there are about 300 elderly migrants over 55 years old living in Community-L, which including two types of more active and more disabled. The first one’s main daily activities are walking, chatting, and working in the community, and their need is more comprehensive with medical care, economy, entertainment, dignity, and value realization. The second one’s main activities are limited to the indoor space due to the limitations of cognitive and activity ability [36,37]. Because this research is aimed at a more comprehensive survey of production and living needs at community level, the more disabled elderly are excluded, and the more active elderly who can independently active in public space are included. The survey was conducted in a working day and a rest day in early July 2018, when the weather conditions were suitable for outdoor activities for the elderly, by a 12-person research team. The investigator randomly approached the elderly by walking within a designated area. Due to elderly migrants’ low overall literacy and low prevalence of Mandarin in Town-D, any respondent who failed to understand the questionnaire after repeated communication was eliminated by the investigator on the spot. Among the 120 questionnaires collected, 4 questionnaires with incomplete information were removed and 116 questionnaires were obtained. Table 2 displays distribution of sample population attributes.

### 2.3. Analytical Framework

The study begins with frequency distribution of the 116 samples’ average scores for production—and living-needs characteristics (Figure 1 and Figure 2). The average score represents the sample’s urgency for the needs. Living needs reflects a low-need sample set within the range of 1.8–2.3 and a high-need sample set within the range of 2.4–3.9. Production needs reflects a low-need sample set within the range of 1–1.5 and a high-need sample set within the range of 1.6–3.2. Since a single indicator’s average score cannot describe the sample’s pattern of urbanization characteristics, the study requires still more in-depth statistical analysis.

The distribution based on the average score of one sample suggests that there are two categories of high need and low need both in production and living dimensions, which means it is necessary to classify and discuss the sample. To explore further comprehensive differences in production- and living-needs indicators among urbanization samples, the study designed an analytical framework, with calculation steps of hierarchical cluster analysis, one-way ANOVA test and Chi-square test performed by SPSS 23.0 (IBM Corporation, Armonk, NY, USA) for Windows and calculation results judged statistically significant at *p* < 0.05:Using hierarchical cluster analysis, which can classify samples according to the closeness degree without prior knowledge; with squared Euclidean distance and between-groups linkage, samples were respectively divided into two groups according to production and living need indicators. Taking the one-way ANOVA test, which can find whether the different levels of a control variable have a significant impact on the observed variable, to find need indicators with significant differences; comparing the average scores of the indicators with significant differences to determine which group is urbanized respectively in living and production dimension, and to assign urbanized values to the samples.According to living and production urbanized values, the study constructed a four-quadrant model frame on human urbanization feature as follows: an Urban Group with both living and production needs urbanized (Group H-H); a Half-urban-half-rural Group with only living needs urbanized(Group H-L); a Half-urban-half-rural Group with only production needs urbanized (Group L-H); and a Rural group with both living and production needs not urbanized (Group L-L) (Figure 3). Each group’ final need indicators are determined according to its average of every need score, with which a need-based urbanization decomposition model is established.Using the chi-square test, which can find the difference between the actual value and theoretical value of an observed variable for categorical control, the study compares the categorical percentage of individual characteristics with the overall percentage, and mines the significant difference of individual characteristics for four urbanization groups. Each group’ final individual characteristics are determined according to every individual percentage, with which an individual-characteristics-based urbanization decomposition model is established.

## 3. Results

### 3.1. Need-Based Classification of Urbanization Characteristics

Samples were classified into Group 1 with 85 samples and Group 2 with 31 samples according to living-needs indicators by hierarchical cluster analysis, and Table 3 displays one-way ANOVA analysis results of two groups: (1) In the six indicators of Community clinics, Community public service center, Card room, Cultural hall, Paths between houses, and Sidewalk, there is little difference in average score of between group 1 and group 2 (*p* > 0.05). (2) In the ten indicators of Community expert medical center, Community canteen, Public bathroom, Day break space, Night rest space, Dancing room, Video room, Study space, park, and small garden, there is significant difference between the two groups (*p* < 0.05). (3) By comparing the overall standard deviation with the group standard deviation in indicators with significant difference, it is found that the standard deviation of group 2 is greatly lower, and group 1 is only slightly higher than the overall in Night rest space and Study space, which indicates that the samples with similar living need are classified and the classification is relatively reasonable. (4) Comparing the average scores of indicators with significant differences, it is found that there is no significant difference for rural characteristics between the two groups, while group 1 has higher need for urban characteristics indexes, indicating that the degree of urbanization of group 1 is higher than that of group 2. Therefore, the group 1 is defined as living urbanized group, group 2 is defined as living unurbanized. (5) The sample size shows that under the influence of urban environment, most people’s living needs tend to be urban characteristics.

According to production-need indicators’ classification, samples are classified into group 1′ with 58 samples and 2′ with 58 samples by hierarchical cluster analysis, and Table 4 lists the one-way ANOVA analysis results of two groups: (1) In the three indicators of Family workshop, Street-facing stores and Labor employment, there is little difference in the average score of between group 1′ and group 2′ (*p* > 0.05). (2) In the six indicators of Farming, Parking lot for agricultural vehicles, Storage space for fertilizer and pesticide, Food handling site, Community vegetable garden, and Vegetable stand, there is a significant difference in the average scores between the two groups (*p* < 0.05). (3) By comparing the overall standard deviation and the group standard deviation in production need indicators with significant difference, it is found that the standard deviation of group 1′ is greatly lower and group 2′ is only slightly higher than the overall in Farming, Food handling site and Community vegetable garden, which indicates that the samples with similar living needs are classified and the classification is relatively reasonable. (4) Comparing the average scores of group 1′ and group 2′, there is no significant difference in Handmade and Business needs between them, while group 1′ has a lower need for agricultural production, indicating that group 1′ is more separated from agriculture, and has a higher production urbanization degree. Therefore, group 1′ is defined as the production urbanized group, and group 2′ is defined as production unurbanized. (5) The sample size shows that, under the influence of an urban environment, there are still a considerable number of humans whose needs remain as rural characteristics.

By permutation and combination, each sample obtained two urbanization values of production and living, and generated four urbanization feature groups. The specific need judgment criteria of a group is that the average score of a certain need of the group is higher than the average score for all production and living needs (2.45). The need-based urbanization feature decomposition model is shown in Figure 4.

**Type 1:** Group 1 and Group 1′. This type represents the urban group urbanized in both production and living with strong environmental adaptability (n = 38), which is defined as Group H-H. The needs for this type include most of the urban public services indicators, all rural public services indicators and some agricultural elements.

**Type 2:** Group 1 and Group 2′. This type represents the half-urban-half-rural group only urbanized in living with moderate environmental adaptability (n = 47), which is defined as Group H-L. The needs for this type include most urban public services indicators, all rural public services indicators and more agricultural elements.

**Type 3:** Group 2 and Group 1′. This type represents the half-urban-half-rural group only urbanized in production with moderate environmental adaptability (n = 20), which is defined as Group L-H. The needs for this type include basic urban-characteristic public services indicators and all rural-characteristic public services indicators. There is no production need for this type.

**Type 4:** Group 2 and Group 2′. This type represents the rural group without urbanized production and living with weak environmental adaptability (n = 11), which is defined as Group L-L. The needs for this type include basic urban public services indicators, all rural public services indicators and more agricultural elements.

### 3.2. Urbanization Types’ Characteristics of Environmental Adaptability

Based on the urbanization classification of population, this study conducted Chi-square analysis on the proportion of each dimension population attribute of four types to understand the impact of different population attributes on urban environmental adaptability (Table 5). (1) The seven population attributes of Gender, Age, Marital status, Housing condition, Physical condition, Work capacity and Educational level are significantly different (*p* < 0.05), which indicates that these seven dimensions have an impact on urban environment adaptability. (2) In Gender, the percentage of Group L-H and Group L-L are greatly different from the whole sample, which indicates that Gender has a great impact on living urbanization; in Age, Work capacity and Educational level, all four groups are very different from the overall proportion, indicating that these three dimensions have an impact on both living and production environmental adaptation; Group L-H’’s low Physical condition percentage shows that the decline of health reduced environmental adaptability; the high proportion of widows in Group H-H and Group L-H indicates that Marital status will affect the degree of production urbanization. (3) For pension, in spite of the little difference among the four groups (*p* > 0.05), we cannot be sure pension has no impact on urban environmental adaptability because the samples in this research are all in low pension. Regardless, the pension standard here and a sample with higher pensions may have different need characteristics.

In population attribute dimensions with significant differences, the percentage of a certain population attribute in one group being higher than the total proportion of the item is used to judge the population’s attribute characteristics of the group. The population-attribute-based urbanization feature decomposition model is shown in Figure 5:

Group H-H, an urban group urbanized in both production and living with strong environmental adaptability: the proportion of males is a little higher, the age is older, the health condition is better, most of them have an independent house, the widow proportion is significantly higher, and work capacity and educational levels are both middling;

Group H-L, a half-urban-half-rural group only urbanized in living with middle environmental adaptability: the proportion of males is a little higher, the age is younger, and their health condition is better; most of them have an independent house, and most still have a spouse, with an overall strong work capacity and high education levels;

Group L-H, a half-urban-half-rural group only urbanized in production with moderate environmental adaptability: the proportion of women is greatly high, with more middle and old aged samples, the proportion of those unable to take care of themselves is high, 40% do not have their own property, 30% are widowed, and there is an overall low work capacity and low educational levels;

Group L-L, a rural group without urbanized production and living with weak environmental adaptability: the proportion of women is very high, with more middle and low aged samples, the health condition is good, more than 60% do not have their own property, most of them have still living spouses, and are mainly engaged in housework, with low educational levels.

### 3.3. Correlation between Need and Environmental Adaptability

Through the qualitative comparison of the needs and the population attributes of four groups with different environmental adaptability, this study found that the urban environmental adaptability of elderly migrants is complicated: one factor is that the production and living needs of different urban environmental adaptability groups is different; the other is that the population attributes of four groups with different environmental adaptability is different. Distribution of 116 samples in a four-quadrant model of human urbanization features shows the relationship of living and production needs and population attributes (Figure 6): Group H-H need most urban public services, all rural public services and some agricultural elements, and tends to be older, in better health, with independent houses, widows, and middling work capacity and educational levels; Group H-L needs most urban public services, all rural public services and more agricultural elements, and tends to be younger, in better health, with an independent house, with a spouse, and with strong work capacity and high education levels; Group L-H needs basic urban public services and all rural public services, and tends to be women, older, unable to take care of themselves, with no independent property, widowed, and with low work capacity and low educational levels; Group L-L needs basic urban public services, all rural public services and more agricultural elements, and tends to be women, low age, in better health, with no property, with spouses still living, mainly engaged in housework, and with low educational levels.

## 4. Discussion

In summary, elderly migrants have the following four characteristics for the production and living needs of resettlement community: (1) in terms of urban public services, basic services were needed by all groups and more were needed by Group H-H and H-L; with the impact of the urbanized environment, urban public services will be more demanded; (2) in terms of rural public services, the four groups have high demand; (3) in terms of Agriculture elements, only the L-H group has no demand; (4) four groups all have no Handmade and Business. Therefore, the production and living needs of elderly migrants are summarized as urban public services, rural public services and agricultural production factors. To meet elderly migrants’ complex needs for production and living, this paper proposes three concepts focused on planning resettlement communities: promotion of urban public services, construction of spiritual homes, and placement of agricultural production factors.

### 4.1. Promote the Supply Level of Urban Public Services

To meet elderly migrants’ increasing need for a better life, resettlement community planning should improve the supply level of urban public services, mainly from two perspectives: (1) ensuring equalization of basic services and (2) enriching diversification of expanded services. Basic services relate to basic living and development; they should include material elements such as housing and municipal facilities, as well as soft elements such as medical care, education, and a safe social environment. To ensure migrants a fair opportunity for basic services, government departments should equalize the provision of public services and creation of the public environment. Expanded services meet needs for improvement of quality of life, involving professional medical care, facilities for the elderly, culture, and entertainment. Expanded services should be diverse both in content and level because of migrants’ differences, and they can operate through introduction of commercial facilities.

### 4.2. Create a Place That Embodies the Spirit of Immigrants’ Homeland

Elderly migrants’ spiritual comfort is mainly realized through ceremonial activities such as festivals, birthdays, weddings, and funerals, along with daily neighborhood interaction activities such as gatherings and extrapolated production and living behaviors. To respect and retain migrants’ traditional memories, reshape their social life, and form good social relations after resettlement, community planning should preserve spaces that reflect the spirit of migrants’ homelands, especially by relying on both collective and neighborhood spaces [37,38,39]. In a narrow sense, collective space generally refers to a cultural hall for such functions as village history shows, banquets, and collective material storage. In a broader sense, collective space also includes public spaces that support a variety of activities and are rich in the sense of domain. Formation of such spaces can generally be achieved by enhancing community boundaries and creating community entrances. Neighborhood spaces are mainly formed by creating small-scale locations with a certain sense of enclosure through terrain height differences, green belts, landscape stones, pavilions, and other elements in spaces between houses, at street intersections, at entrances and exits of residential groups, and around stores of specific business types.

### 4.3. Moderate Consideration of Agricultural Production Needs

At the community level, provision of agricultural production factors mainly consists of the installation of a certain percentage of green space for planting and construction of production service facilities [40,41]. Small-scale planting sites can be established in combination with green spaces between houses and alongside public facilities [42]. For needs of agricultural vehicle parking, fertilizer and pesticide storage, and agricultural operations, production service facilities can be provided in conjunction with community service centers and parks, while demand for agricultural technology services can be directed to commercial development. In terms of industry and commerce, although elderly migrants’ needs were not high, in interviews, they expressed that the main obstacle to engaging in such production methods was their lack of job skills and opportunities. With elderly migrants’ gradual urbanization, the number of those with the skills to engage in industry and commerce will gradually increase, and resettlement community planning should consider such needs.

## 5. Conclusions

The resettlement community has achieved environmental urbanization in terms of natural land, planning mode, spatial pattern, and preset production methods. However, elderly migrants’ production and living habits continue rural traditions, in contrast to the urban residential environment. This reflects a problem in that human and environmental urbanization are not synchronized.

Based on the internal—human—factor of environmental regulation, this study begins with individual needs of elderly migrants, and attends to interaction mechanisms between the environment and human beings. Then, the study employs the systematic clustering method to construct a human urbanization feature, a four-quadrant model according to people’s production and living needs. Through the qualitative comparison the needs and the population attributes of four groups with different environmental adaptability, the study revealed four features: Group H-H needs most urban public services, all rural public services and some agricultural elements, and tends to be older, in better health, with independent houses, widows, and has middling work capacity and educational levels; Group H-L needs most urban public services, all rural public services and more agricultural elements, and tends to be younger, in better health, with independent houses, with spouses, and with strong work capacity and high education levels; Group L-H needs basic urban public services and all rural public services, and tends to be women, older, unable to take care of themselves, with no property, widowed, and with low work capacity and low educational levels; and Group L-L needs basic urban public services, all rural public services and more agricultural elements, and tends to be women, low age, in better health, with no property, with spouses still living, mainly engaged in housework, and with low educational levels.

With regard to elderly migrants’ urbanization cognition of living environment, this study focuses only on the functional plan of production and living needs. However, such urbanization cognition was also reflected in elderly migrants’ preference for spatial characteristics (e.g., spatial domain, accessibility, identifiability). Future research may focus on these indicators. In addition, due to limitations of time and resources, this study investigated only one resettlement community. Other resettlement communities might manifest different distribution characteristics in the two-dimensional model of production and living; correlation between needs and environmental adaptability might also differ from this case. Additionally, features in this case’s model need more verification. With gradual urbanization, elderly migrants’ dynamic needs should also be considered in future research.

## Figures and Tables

**Figure 1 ijerph-18-05068-f001:**
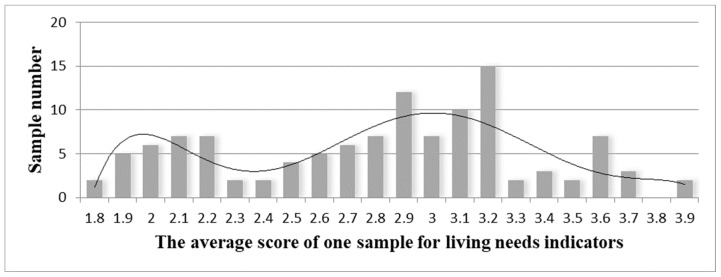
Frequency distribution of the average score of one sample for living needs.

**Figure 2 ijerph-18-05068-f002:**
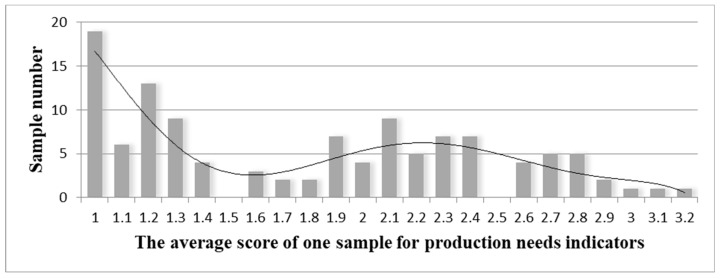
Frequency distribution based on the average score of one sample for production needs.

**Figure 3 ijerph-18-05068-f003:**
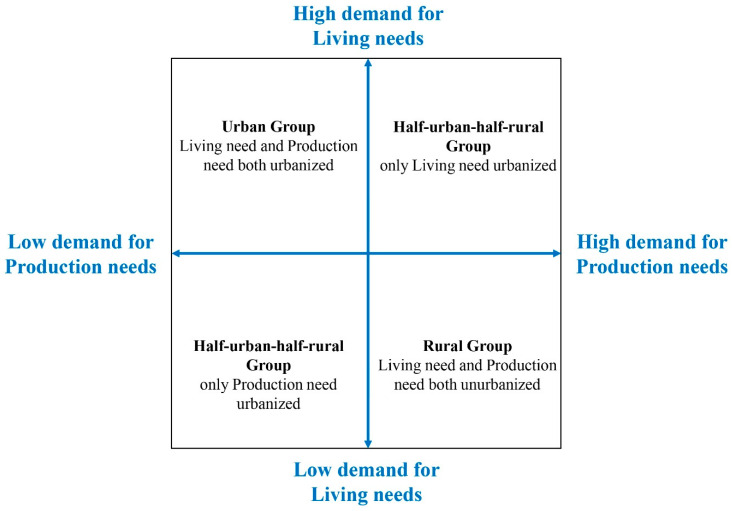
Framework of the model of urbanization characteristics (Source: Author).

**Figure 4 ijerph-18-05068-f004:**
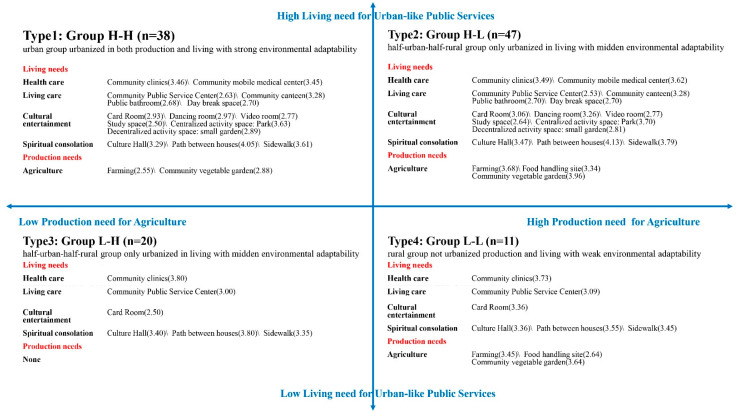
Decomposition model of urbanization features based on need (Source: Author).

**Figure 5 ijerph-18-05068-f005:**
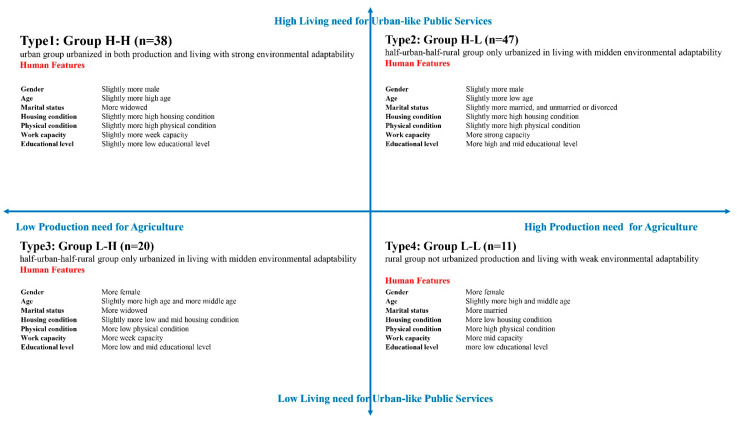
Decomposition model of urbanization features based on population-attribute (Source: Author).

**Figure 6 ijerph-18-05068-f006:**
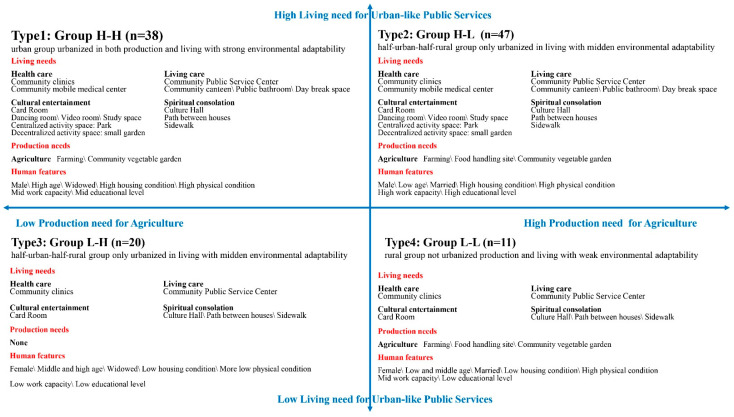
Model of urbanization characteristics (Source: Author).

**Table 1 ijerph-18-05068-t001:** Production- and living-needs indicator system.

First-Level Indicator	Second-Level Indicator	Third-Level Indicator
Living Needs	Health care	Community clinics
		Community mobile medical center
	Living care	Community public service center
		Community canteen
		Public bathroom
		Day break space
		Overnight restroom
	Cultural entertainment	Card room
		Dancing room
		Video room
		Study space
		Centralized activity space: Park
		Decentralized activity space: small garden
	Spiritual consolation	Culture hall
		Path between houses
		Sidewalk
Production Needs	Agriculture	Farmland
		Parking for agricultural vehicles
		Storage space for fertilizer and pesticide
		Farm operation space
		Community agricultural garden
		Vegetable stall
	Handmade and Business	Family workshop
		Street shop
		Labor employment

**Table 2 ijerph-18-05068-t002:** Basic demographic information and population attributes of the survey sample (N = 116).

Name	Option	Explanation	Frequency	Percent %
Gender	Female	/	64	55.17
	Male	/	52	44.83
Age	Low	55–69 years	49	42.24
	Mid	70–79 years	46	39.66
	High	Older than 80 years	21	18.1
Marital status	Married	/	87	75
	Widowed	/	26	22.41
	Unmarried, divorced	/	3	2.59
Pension	Low	Less than 1000 ¥	63	54.31
	Mid	1000–2000 ¥	33	28.45
	High	More than 2000 ¥	20	17.24
Housing condition	Low	Living with other family members, no house ownership	14	12.07
	Mid	Living alone, no house ownership	14	12.07
	High	House ownership	88	75.86
Physical condition	Low	Needing help or nursing	7	6.03
	High	Self-caring	109	93.97
Work capacity	Weak	Can neither undertake housework nor make money	27	23.28
	Mid	Mainly housework	62	53.45
	Strong	Can make money	27	23.28
Educational level	Low	Not completed primary education	59	50.86
	Mid	Completed primary education	42	36.21
	High	Completed junior school education and above	15	12.93

**Table 3 ijerph-18-05068-t003:** Characteristic variables of living needs and One-Way ANOVA test (Source: Author).

Living Demand Indicators	Classification of Living Needs (Mean ± Standard Deviation)		Average Score (Mean ± Standard Deviation)	*p*
	1 (n = 85)	2 (n = 31)		
Community clinics	3.49 ± 0.84	3.77 ± 0.92	3.57 ± 0.87	0.124
Community experts medical center	3.48 ± 0.96	1.87 ± 0.62	3.05 ± 1.13	0.000 **
Community public service center	2.73 ± 0.90	2.87 ± 0.85	2.75 ± 0.93	0.450
Community canteen	3.31 ± 0.95	1.52 ± 0.72	2.83 ± 1.20	0.000 **
Public bathroom	2.69 ± 0.93	1.19 ± 0.40	2.30 ± 1.06	0.000 **
Day break space	2.72 ± 1.11	1.13 ± 0.56	2.29 ± 1.22	0.000 **
Night rest space	2.19 ± 1.20	1.03 ± 0.18	1.88 ± 1.15	0.000 **
Card room	2.95 ± 1.05	2.81 ± 1.22	2.91 ± 1.09	0.525
Dancing room	2.99 ± 1.32	1.03 ± 0.18	2.47 ± 1.43	0.000 **
Video room	2.79 ± 1.37	1.13 ± 0.43	2.34 ± 1.40	0.000 **
Study space	2.52 ± 1.29	1.16 ± 0.45	2.16 ± 1.28	0.000 **
Centralized activity space: park	3.66 ± 1.26	2.16 ± 0.97	3.26 ± 1.36	0.000 **
Decentralized activity space: small garden	2.91 ± 1.20	1.16 ± 0.37	2.44 ± 1.30	0.000 **
Cultural hall	3.32 ± 0.89	3.39 ± 1.12	3.34 ± 0.95	0.729
Paths between houses	4.08 ± 1.21	3.71 ± 0.82	3.97 ± 1.14	0.115
Sidewalk	3.64 ± 1.23	3.39 ± 0.76	3.58 ± 1.17	0.297

** *p* < 0.01.

**Table 4 ijerph-18-05068-t004:** Characteristic variables of production needs and One-Way ANOVA test (Source: Author).

Production Demand Indicators	Classification of Production Needs (Mean ± Standard Deviation)		Average Score (Mean ± Standard Deviation)	*p*
	1′ (n = 58)	2′ (n = 58)		
Farming	1.12 ± 0.38	3.64 ± 1.02	2.38 ± 1.47	0.000 **
Parking lot for agricultural vehicles	1.09 ± 0.28	2.03 ± 0.73	1.56 ± 0.72	0.000 **
Storage space for fertilizer and pesticide	1.05 ± 0.29	2.21 ± 1.02	1.63 ± 0.94	0.000 **
Food handling site	1.14 ± 0.44	3.21 ± 1.53	2.17 ± 1.52	0.000 **
Community vegetable garden	1.55 ± 0.92	3.90 ± 1.00	2.72 ± 1.51	0.000 **
Vegetable stand	1.16 ± 0.45	1.98 ± 1.07	1.57 ± 0.91	0.000 **
Family workshop	1.29 ± 0.56	1.57 ± 0.90	1.43 ± 0.76	0.050
Street-facing stores	1.19 ± 0.69	1.38 ± 0.67	1.28 ± 0.68	0.135
Labor employment	1.50 ± 0.82	1.31 ± 0.63	1.41 ± 0.73	0.165

** *p* < 0.01.

**Table 5 ijerph-18-05068-t005:** Results of cross-sectional (chi-square) analysis of urbanization types and elderly migrants’ individual characteristics (Source: Author).

Name	Option	Percentage of Type of Urbanization (N)				Average Percentage	χ²	*p*
		Type 1 (38) Urban Group Urbanized Both in Living and Production	Type 2 (47) Half-Urban-Half-Rural Group ONLY Urbanized in Living	Type 3 (20) Half-Urban-Half-Rural Group Only Urbanized in Production	Type 4 (11) Rural Group Not Urbanized Both in Living and Production			
Gender	Female	44.7%	48.9%	80.0%	75.0%	55.2%	8.768	0.033 *
	Male	55.3%	51.1%	20.0%	25.0%	44.8%		
Age	Low	29.0%	63.8%	20.0%	36.4%	42.2%	26.108	0.000 **
	Mid	36.8%	31.9%	50.0%	63.6%	39.7%		
	High	34.2%	4.3%	30.0%	0.0%	18.1%		
Marital status	Married	63.2%	83.0%	70.0%	90.9%	75.0%	13.773	0.032 *
	Widowed	36.8%	10.6%	30.0%	9.1%	22.4%		
	Unmarried, divorced	0.0%	6.4%	0.00%	0.0%	2.6%		
Pension	Low	60.5%	48.9%	65.0%	36.4%	54.3%	4.232	0.645
	Mid	23.7%	29.80%	25.0%	45.5%	28.5%		
	High	15.80%	21.3%	10.0%	18.2%	17.2%		
Housing condition	Low	10.5%	6.4%	25.0%	18.2%	12.1%	21.474	0.002 **
	Mid	2.6%	10.6%	15.0%	45.5%	12.1%		
	High	86.8%	83.0%	60.0%	36.4%	75.9%		
Physical condition	High	97.4%	97.90%	75.0%	100%	94.0%	15.434	0.001 **
	Low	2.6%	2.1%	25.0%	0.0%	6.0%		
Work capacity	Weak	29.0%	8.5%	45.0%	27.3%	23.3%	18.059	0.006 **
	Mid	44.7%	57.5%	55.0%	63.6%	53.5%		
	Strong	26.3%	34.0%	0.0%	9.1%	23.3%		
Educational level	Low	52.6%	27.7%	85.0%	81.8%	50.9%	33.864	0.000 **
	Mid	44.7%	42.6%	15.0%	18.2%	36.2%		
	High	2.6%	29.8%	0.0%	0.0%	12.9%		

* *p* < 0.05, ***p* < 0.01.

## Data Availability

Data supporting reported results can be found in https://pan.zju.edu.cn/share/1e37d0e152c8a7e624edeab492 (Accessed on 10 May 2021).

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
