# Peer review of "Correlation between Elderly Migrants’ Needs and Environmental Adaptability: A Discussion Based on Human Urbanization Features"

_ijerph, 2021, doi:10.3390/ijerph18105068_

Round 1

Reviewer 1 Report

The paper is interesting. Its organization with sections (Introduction, Literature Review and Research Methods, Results, Discussion, Conclusions) is adequate, the material is ordered in a way that is logical, clear, and easy to follow. The authors cited sources adequately and appropriately, and all the citations in the text are listed in the References section.

Research data are presented and visualized in a clear way, which all makes the paper readable. The methods are described in a proper way. The paper is well structured and well written.

Author Response

Dear reviewer:

Thank you for your time and attention during your busy schedule. We are really grateful for your appreciation and recognition of our papers. And looking forward to having the opportunity to have in-depth academic exchanges with you.

Thank you again!

Reviewer 2 Report

Very interesting research topic even if it conducted only with statistical tools that often do not take into account human variables that are very difficult to schematize with numbers and formulas.

Precisely for this reason, the study deserves a greater study with respect to the history of elderly migrants, their origin and therefore their original culture and customs.

In this way you can better understand their needs and therefore how to meet them.

Arrange table 1 so that you understand the sublevels to which higher level they correspond.

Improve the readability of figures 1 and 2 since the correspondence with the indicators of table 1 is not immediate

Author Response

Thank you for your time and attention during your busy schedule. We are really grateful for your comments. The responses are briefly described in the  attachment. Besides, the editorial corrections and the changes are marked with red in the revised manuscript. 

Reviewer 3 Report

The paper entitled "Correlation Between Elderly Migrants' Needs and Environmental Adaptability: A Discussion Based on Human Urbanization Features" studies elderly migrants' needs, exploring interaction mechanisms between the enivironment and humans. The study uses the systematic clustering approach to construct a four-quadrant model according to production and living needs, and a chi-square analysis to explore correlation between environmental adaptability and needs. The methodology followed in the paper seems sound and the entire analysis and its results are interesting and comprehensible. However, the quality of the paper could be enhanced including the following comments:

1) In Section 2.2. of the paper the authors should include the procedure or formula utilized to select the appropriate sample size for their analysis. Appropriate references could be included instead.

2) In Section 2.2. of the paper the authors could include a short description of both the one-way ANOVA and the chi-square analysis, so that the reader can more easily perceive the results extracted.

3) In Section 3 of the paper the authors should try to include more insight on extracted results presented in Tables 4, 5 and 6. They should try to comment further on results of classification of living needs and production urbanization, considering also statitical significance of the results. They should also try to comment further on results of Table 6, based on estimates of chi-square statistics and its associated p-values.

4) In Section 3.3. of the paper the correlation between need and environmental adaptability is considered. Correlation can be used to quantify the linear dependency of two variables. It cannot capture non-linear relationship between variables. Was non-linear dependency assessed in this paper, and if it has been considered how? What is the correlation measure used in this work?

Some minor comments are also given below:

1) In Section 2.2., references could be provided for the Reservoir-L project, as well as for the Likert scale used.

2) Table 2 could be more easy to read if blank lines were included between investigators.

3) In Section 2.2. it is better not to refer to the software used (SPSS 23.0 for Windows), but instead focus on the methodology utilized.

4) In Table 4 why is the first result underlined? The same also holds for Table 6.

5) In Table 4, the exponents of F and p are difficult to read.

6) Section 3.2 - Line 1- "Different types of ... in Table 6:"

7) Section 3.3. - Line 3 refers to Figure 4, but there exists a Figure 5 in the paper.

Author Response

Thank you for your time and attention during your busy schedule. We are really grateful for your comments. The responses are briefly described in the attachment . Besides, the editorial corrections and the changes are marked with red in the revised manuscript.

Round 2

Reviewer 2 Report

I understand the difficulty of integrating the contents with the part I had suggested, however the article works and can be published in this latest version